# Adaptive Probing Policies for Shortest Path Routing

**Aditya Bhaskara**
School of Computing
University of Utah
bhaskaraaditya@gmail.com

**Sreenivas Gollapudi**
Google Research
Mountain View, CA
sgollapu@google.com

**Kostas Kollias**
Google Research
Mountain View, CA
kostaskollias@google.com

**Kamesh Munagala**
Department of Computer Science
Duke University
kamesh@cs.duke.edu

## Abstract

Inspired by traffic routing applications, we consider the problem of finding the shortest path from a source $s$ to a destination $t$ in a graph, when the lengths of the edges are unknown. Instead, we are given *hints* or predictions of the edge lengths from a collection of ML models, trained possibly on historical data and other contexts in the network. Additionally, we assume that the true length of any candidate path can be obtained by *probing* an up-to-date snapshot of the network. However, each probe introduces a latency, and thus the goal is to minimize the number of probes while finding a near-optimal path with high probability. We formalize this problem and show assumptions under which it admits to efficient approximation algorithms. We verify these assumptions and validate the performance of our algorithms on real data.

## 1 Introduction

Routing traffic is a prototypical example of using large scale ML for finding shortest paths in graphs where the state of the graph is constantly changing with time. Given the scale of the road network with billions of road segments around the world [Strano et al., 2017], and low latency requirements for the path search algorithms [Goldberg, 2007, Min and Wynter, 2011], integrating ML models that compute edge or path lengths into the path search algorithms is a challenging problem. Further, the architecture to serve route recommendations to users needs to deal with network data at different levels of granularity [Baum et al., 2016, Delling et al., 2017, 2018].

Typically, the lengths, either for edges or paths, are computed by multiple ML models which use some combination of historical network statistics as well as current state of the network [Yang et al., 2004, de Fabritiis et al., 2008, Tchrakian et al., 2012]. These predicted lengths are then consumed by the path searcher to generate route recommendations. The details of the ML models are opaque to this path searcher. To handle any errors in the predicted lengths gracefully, the path searcher queries a real-time traffic source that keeps a more accurate representation of the network [Cebecauer et al., 2018] and gets updated at a much higher frequency and possibly on a much smaller fraction of the network than the ML predictions. However, access to the traffic server is expensive: it consumes critical time that affects the latency of serving the user request. Therefore, trading off between the number of probes to the traffic server and error in the predicted lengths becomes an important design decision in engineering an efficient and effective routing engine. Figure 1 summarizes the architecture.

In this paper, we study the setting where a path searcher has access to multiple predictions for path lengths, potentially from multiple ML models. Using these predictions or *hints*, the searcher can query or probe a traffic server to get the accurate length for any path. The goal is to compute the shortest path between two end points with high probability using all the hints and a small a number of probes to the real-time traffic server.

## 1.1    Model and Results

For our theoretical results, we consider the problem of routing from a source $s$ to sink $t$ in a specialized network composed of $n$ parallel edges between $s$ and $t$, each edge of different length. Though this problem appears stylized, as we discuss below, our main empirical contribution is to show that the theory we develop for this problem can be applied to routing in real-world networks.

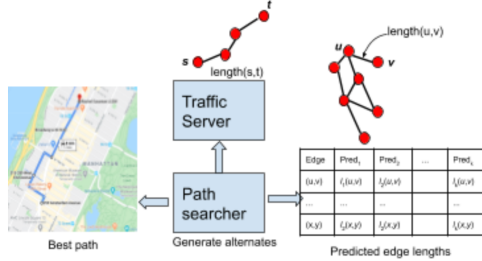

Figure 1: Architecture of a routing engine

Going back to our stylized network, the true length $L_j$ of any edge $j$ is unknown upfront. Each of $m$ predictors (or *experts*) makes a prediction for each edge, where $P_{ij}$ denotes the prediction of expert $i$ for edge $j$. We can *probe* any edge $j$ (say via a traffic server) to obtain its true length $L_j$; however, these probes consume server resources and incur latency, and are hence *expensive*. The goal is to devise a probing strategy that unearths the edge with minimum true length with as few probes as possible, with high probability.

**Arbitrary Prediction Errors.** We first consider the model where the predictions of the experts can be arbitrary. Here, we cannot hope to recover the minimum length edge exactly unless we probe all edges. We show that overcoming this barrier requires that our algorithm makes an error that is at least the maximum error made by the best expert, as well as the maximum error on the minimum length edge. On the other hand, we show that such an error can be achieved without performing *any* probes. In essence, the model where experts can make arbitrary errors is too pessimistic and does not offer much insight into algorithm development. For lack of space, we present this result in Appendix A in the supplementary material.

**Stochastic Prediction Errors.** In Section 2, we therefore assume the prediction errors of the experts are drawn randomly from known, independent distributions. In this case, by a simple application of Bayes' rule, the predictions $\{P_{ij}\}$ induce a posterior distribution over the true lengths $\{L_j\}$, where the distributions for each edge are independent. In this case, we are given a bound $\delta$ on the probability that the minimum length edge found by probing is not the overall minimum length edge. We seek to minimize the expected number of probes needed to achieve this bound. Such a probing strategy could be *adaptive*, depending on the outcomes of the edges probed so far.

Our main technical contribution is a reduction of the above problem to adaptive submodular maximization, where the non-trivial aspect is the construction of an appropriate *surrogate* submodular function to capture the outcome of the probing process. Using this reduction, we show a simple greedy policy that probes a factor $O(\log^2(n/\delta))$ more edges in expectation, and achieves a constant approximation to the probability $\delta$ that the chosen edge is not the overall best edge.

**Empirical Validation.** Our theoretical model assumes parallel edges whose lengths follow independent distributions. We can port such a model to real traffic networks where each $s$–$t$ path in the traffic network maps to one of the parallel edges. The main issue that arises in path routing is that paths can overlap, and hence the path lengths can no longer be treated as independent random variables.

In Section 3, we empirically demonstrate how to overcome these obstacles on real-world data, and reduce the overall probing problem to the parallel edge case. Towards this end, we analyze data from the NYC traffic network. We show that conditioned on knowing the length of one canonical path, the *fluctuations* in the lengths of the other paths are approximately independent. This reduces the problem subsequent to probing the canonical path to the parallel edge case. We simulate the greedy algorithm for the parallel edge case on our path data under the independence assumption, and show

that with a small number of probes we are able to recover the optimal path in almost all instances, validating its performance in practice.

## 1.2 Related Work

**Stochastic Probing.** Stochastic models for probing independent distributions have a rich history in algorithm design, primarily due to applications in database query optimization [Munagala et al., 2005, Deshpande et al., 2016, Liu et al., 2008] and wireless communication [Guha et al., 2006]. In these problems, we are given $n$ independent distributions each of which can be probed at a certain cost, revealing its true value. These problems either have a bound on the number of probes, or seek to minimize the expected probing cost in order to optimize a certain objective over the probed and unprobed values. Of particular relevance is the question of adaptively probing at most $k$ distributions to minimize the smallest value discovered among the probed distributions [Goel et al., 2006]. A related problem is the *Pandora's problem* [Guha et al., 2006, Beyhaghi and Kleinberg, 2019] that seeks to maximize the largest value found minus the total probing cost spent in discovering the value. Typical approaches to solving these problem involve greedy strategies that are based on *submodularity* of objectives such as the maximum of a set of distributions. A general adaptive greedy algorithm for such problems, which probes the next distribution conditioned on the values seen so far, was presented in Golovin and Krause [2011]. The performance guarantee requires submodularity to hold for every realization of probed values so far.

Our work is different from formulations considered in query optimization and wireless communication in that we seek to do more than simply approximate the smallest value found; instead we seek to find the true smallest value (had we probed all the distributions) with high probability. This makes a direct application of adaptive submodularity infeasible, and we make our main technical contribution of using a surrogate submodular function to model our objective.

**Stochastic Shortest Paths.** Our work is also related to the body of literature on *shortest paths under uncertainty*. It is typically assumed that edge lengths follow known independent distributions. A canonical problem is to find an $s$–$t$ path whose length is below a threshold $L$ with highest probability. When the edge length distributions are Gaussian, Nikolova et al. [2006] present a quasi-polynomial time algorithm for this problem via connections to quasi-convex maximization. However, no generalization is known when the distributions are not Gaussian. A related probing problem is the *Canadian Traveler Problem* [Nikolova and Karger, 2008, Papadimitriou and Yannakakis, 1991], where the length of an edge is revealed when we reach one of its end-points. The goal is to find an adaptive routing policy from $s$ to $t$ that has minimum expected length; such a policy could backtrack on edges it has already seen. There are no efficient algorithms known for this problem, except under special assumptions such as no backtracking [Bnaya et al., 2009]. The difficulty is that though edge lengths are independent, the path lengths can be arbitrarily correlated and this can make the problem intractable in the worst case.

In light of this worst-case difficulty, we take a more *data-centric* approach. We show that on realistic instances, the path lengths are roughly independent conditioned on knowing the length of one path. This reduces the problem to a simpler probing problem over independent distributions, and we extend machinery based on adaptive submodular function maximization to solve it.

**Online Problems with Hints.** Adaptive or online algorithms with machine learning predictions has recently been popular for a variety of problem domains. Traditionally, in online algorithms, the future is assumed to be entirely unknown to the algorithm, which often results in pessimistic solutions. In contrast, recent research has focused on incorporating machine learning predictions in online algorithms to obtain more optimistic bounds if the predictions are correct, but preserve the robustness of the classic model in case the predictions turn out to be inaccurate. This model has been applied to a wide variety of problems including rent or buy problems [Purohit et al., 2018, Gollapudi and Panigrahi, 2019], caching [Lykouris and Vassilvtiskii, 2018], scheduling [Lattanzi et al., 2020], frequency estimation [Hsu et al., 2019], Bloom filters [Mitzenmacher, 2018], and so on.

In the context of probing for shortest paths, we assume each machine learning expert makes a prediction for the length of each edge. However, unlike prior work that admits positive results even when experts can make arbitrary errors, we show that if the experts can be arbitrarily inaccurate, the bounds we obtain for shortest paths are still too pessimistic. We therefore need to make stochastic assumptions on the accuracy of the experts themselves. In essence, this reduces to a stochastic

probing problem where the distribution on edge lengths is induced by the noise distributions of the experts, and making any weaker assumptions on the experts leads to pessimistic bounds.

## 2 Stochastic Prediction Error Model

In this section, we consider the following simple, yet canonical, graph model. The graph consists of $n$ parallel paths between $s$ and $t$. We assume that there are $m$ *predictors* (or experts) that predict a length for each of the $n$ edges. $P_{ij}$ denotes the $i$th expert's prediction for the $j$th edge, and the true length of the edge is denoted by $L_j$.

In Appendix A (see supplementary material), we consider the case where we do not have any assumptions on the predictions. In this case, we show lower bounds that make it difficult to obtain non-trivial algorithms. In particular, we present examples illustrating that if the predictions are adversarial, then probing does not offer much advantage.

We now show that if we make stochastic assumptions on the predictions, specifically that a predictor's error on each edge is randomly distributed (according to a known distribution, which can potentially be obtained from historic data), then an adaptive greedy probing strategy is provably effective.

**Stochastic Model.** Our approach has two main steps. The first is to use the assumptions on the predictors' errors to obtain a posterior distribution for the length of each path. Next, we develop a strategy for probing the paths that takes these distributions into account, and aims to maximize the probability of finding a near-minimal path with as few queries as possible.

We assume there is a prior distribution $\mathcal{D}_j$ over the edge length of $j$. Given the true edge length $L_j$, each expert outputs a prediction $P_{ij} = L_j + \eta_{ij}$, where the error $\eta_{ij}$ is drawn from a known independent distribution. Given the predictions $\{P_{ij}\}$, we can use Bayes' rule to compute a distribution $X_j$ over the true length $L_j$ for each edge $j$ as:

$$g_j(L) = \Pr[X_j = L] = \frac{\prod_i \Pr\left[\eta_{ij} = P_{ij} - L\right] \cdot \Pr[\mathcal{D}_j = L]}{\sum_{L'} \prod_i \Pr\left[\eta_{ij} = P_{ij} - L'\right] \cdot \Pr[\mathcal{D}_j = L']}$$

In what follows, we will therefore ignore the predictions, and simply assume access to the conditional distributions $X_j$ for each edge $j$. We denote by $E$ the set of all edges, and assume $|E| = n$.

### 2.1 Adaptive Probing Strategies

We can thus formalize the probing question as follows: we have $n$ paths between $s$ and $t$. The length of path $j$ is a random variable $X_j$ with density function $g_j$. Moreover, the $\{X_j\}_{j=1}^n$ are independent r.v.s. We assume for notational simplicity that $X_j$ are all discrete and have the common support $\Lambda$. As all the paths are independent, we can view them as single edges, without loss of generality. When we probe/query an edge, we observe a realization of $X_j$.

Our goals are the following: (a) make as few probes as possible, (b) maximize the probability of finding an edge that is within $\epsilon$ of $\min_j X_j$, where $\epsilon$ is a given accuracy parameter. The accuracy parameter captures the idea that a small amount of sub-optimality is acceptable in most situations (e.g., a finding a path that takes a few minutes longer than optimal).

Formally, a probing *policy* is described by a tree. At every step, we have a collection of *observations* (values of edges probed so far). Based on these values, a new edge is probed, and we see a realization of the length of that edge. At some point, the policy terminates (stops probing) and outputs the shortest edge seen so far. We say that a policy succeeds with parameters $(\epsilon, \delta)$ if at *any* termination, if we denote by $S$ the set of edges probed so far and by $L$ the minimum probed edge length, we have:

$$\Pr\left[\min_{j \notin S} X_j \leq L - \epsilon\right] \leq \delta \quad \forall (S, L). \tag{1}$$

**Objective.** Given $\epsilon, \delta > 0$, the goal is to find a probing policy $\pi$ that succeeds with parameters $(\epsilon, \delta)$, such that the *expected number of probes* is minimized. The expectation is over the realizations of the values of $X_j$. Our main result is the following:

**Theorem 2.1.** *Let $\epsilon, \delta > 0$ be given parameters, and suppose $X_j$ are independent random variables whose distributions $g_j$ are known. Suppose there exists a probing strategy $\pi^*$ that succeeds with parameters $(\epsilon, \delta)$ and makes OPT probes in expectation. Then the adaptive greedy strategy (Algorithm 1) succeeds with parameters $(\epsilon, 3\delta)$ and makes $O(\text{OPT} \cdot \log(n/\delta))$ probes in expectation.*

## 2.2   Background: Adaptive Submodularity

We use the *adaptive submodularity* framework of Golovin and Krause [2011]. We begin with some notation and definitions. Let $E = \{e_1, e_2, \ldots, e_n\}$ denote the $n$ edges. Recall that $X_j$ is the random variable denoting the length of $e_j$. It has the density function $g_j$.

**Definition 2.2** (Golovin and Krause [2011])**.** *Let us define the following terms:*

- *A* realization *$\phi$ is an assignment of lengths to all the edges. Specifically, $\phi : E \mapsto \mathbb{R}$, and the probability of this realization (since $X_j$ are independent) is $\prod_j \Pr[X_j = \phi(e_j)]$.*

- *An* observation *(or partial realization) $\psi$ consists of a subset $S$ of the edges along with their realized lengths. $S$ is called the* domain *of $\psi$. Formally, we view $\psi$ as a set of pairs $(e, \ell)$, for $e \in S$ (there is exactly one pair for each $e \in S$). Since only the smallest observed matters for many of our arguments, we sometimes write $\psi = (S, L)$, where $S$ is the queried set, and $L$ is the minimum observed length.*

- *We say that an observation $\psi$ is* consistent *with a realization $\phi$ if $\phi(e) = \ell$ for all pairs $(e, \ell) \in \psi$. We write this as $\psi \sim \phi$.*

- *For two observations $\psi = (S, L), \psi' = (S', L')$, we say that $\psi \preceq \psi'$ if every pair $(e, \ell)$ in $\psi$ is also in $\psi'$. This implies that $S \subseteq S'$ and $L \geq L'$.*

The main quantity of interest is the function $f : 2^E \times \Lambda^E \mapsto \mathbb{R}$ (recall that $\Lambda$ is the set of all possible edge lengths). Suppose we probed a subset $S$ of edges, $f(S, \phi)$ denotes a "utility" we associate with the probes $S$ for the realization $\phi$. In our application, the value of $f(S, \phi)$ will only depend on $S$ and the values $\phi(S)$; i.e., they do not depend on the lengths of the edges $E \setminus S$. Such a function satisfies the so-called *self-certifying* property, defined in Golovin and Krause [2011]. Formally, if we have a realization $\phi$ and an observation $\psi = (S, L)$ that is consistent, $f$ is said to have the self-certifying property if $f(S, \phi) = f(S, \phi')$ for all other realizations $\phi'$ such that $\psi \sim \phi'$. This property holds, in particular, if $f$ only depends on $\psi$ (as will be the case for us).

We consider functions $f$ where for all realizations $\phi$, $f(\emptyset, \phi) = 0$ and $f(E, \phi) = Q$, for some parameter $Q$. In other words, if we make no queries, the utility is 0 and if all edges are queried, the utility is $Q$. The framework of Golovin and Krause [2011] aims to query a small set $S$, while achieving an $f()$ value of $Q$. This turns out to be possible if $f$ satisfies certain structural properties, that we now define.

**Definition 2.3** (Monotonicity)**.** *Let $\psi$ be an observation with domain $S$, and let $e \notin S$ and $\ell \in \Lambda$. Define $\psi' = \psi \cup \{(e, \ell)\}$. $f$ is said to be* strongly adaptive monotone *for all $(e, \ell)$ as above, we have*

$$\mathbb{E}_\phi[f(S, \phi) \mid \psi \sim \phi] \leq \mathbb{E}_\phi[f(S \cup \{e\}, \phi) \mid \psi' \sim \phi].$$

The expectations above are over all the $\phi$ that are consistent with the corresponding $\psi$. In the case when $f$ is only dependent on $\psi$ (as will be the case for us), the above is equivalent to $f(\psi) \leq f(\psi')$.

A second property we need is related to submodularity. Before defining this, let us introduce another notation. Let $\psi$ and $\psi'$ be observations with domains $S$ and $S'$ respectively, and let $\psi \preceq \psi'$. Let $e \in E \setminus S'$. Define

$$\Delta(e|\psi; \psi') := \mathbb{E}_\phi\big[f(S \cup \{e\}, \phi) - f(S, \phi) \mid \psi' \sim \phi\big].$$

The expectation runs over all $\phi$ consistent with $\psi'$. Since $e \notin S'$, the expectation runs over all the realized lengths of the edge $e$. We can now define strong adaptive submodularity as follows:

**Definition 2.4.** *[Submodularity] A function $f$ is* strongly adaptive submodular *if for all $\psi, \psi', e$ as above, we have $\Delta(e|\psi; \psi') \geq \Delta(e|\psi'; \psi')$. In other words, conditioned on a realization consistent with $\psi'$, the marginal increase in $f$ is at least as large when $e$ is probed in $\psi$ compared to $\psi'$.*

Given the above definitions, the main result in Golovin and Krause [2011] is about the adaptive GREEDY algorithm: At each step, given an observation $\psi = (S, L)$, probe the edge $e \notin S$ that maximizes $\Delta(\psi, e; \psi)$, stopping when $f > Q - \eta$ for suitably chosen $\eta$.

Consider all adaptive policies that only stop when $f > Q - \eta$, and among them, let $\pi^*$ minimize the expected number of probes. Let this optimal expected number of probes be denoted OPT.

**Theorem 2.5** (Golovin and Krause [2011])**.** *Suppose $f$ is self-certifying, strongly adaptive monotone, and strongly adaptive submodular. Further, suppose $\eta$ is such that $f(S, \phi) > Q - \eta$ implies*

$f(S, \phi) = Q$. *Then* GREEDY *has expected number of probes at most* $O\left(\text{OPT} \cdot \left(\ln\left(\frac{Q}{\eta}\right) + 1\right)^2\right)$.

## 2.3  Surrogate Submodular Function

Let us now see how to apply the methods of Golovin and Krause [2011] to our setting. A natural way to define $f$ is as follows: after querying $S$, the probability that none of the non-queried edges has a length significantly smaller than the minimum length of edges in $S$. Formally, $f(S, \phi) = 1 - \Pr[\min_{e_j \notin S} X_j \leq L - \epsilon]$, where $L = \min_{e \in S} \phi(e)$. This function satisfies the self-certifying property, since it does not depend on the realized lengths of edges not in $S$. However, it does not satisfy (strong) adaptive submodularity.

**Example 1.** *Consider two edges $a, b$. Suppose $X_a$ is $1/4$ w.p. $1/2$ and $3/4$ w.p. $1/2$. Suppose $X_b$ is $0$ w.p. $1/2$ and $1/2$ w.p. $1/2$. Now, consider the two observations $\psi = \emptyset$ (no queries) and $\psi' = \{(b, 1/2)\}$ (i.e., $b$ was queried and the observed value is $1/2$). Now, we have $\Delta(a|\psi; \psi') = 1/4$, while $\Delta(a|\psi'; \psi') = 1/2$. This violates Definition 2.4.*

To circumvent this problem, we use a *linear surrogate* function for which all the prerequisites for Theorem 2.5 hold. Recall that $g_j(\ell) = \Pr[X_j = \ell]$ for all $\ell \in \Lambda$, and that $E = \{e_1, e_2, \ldots, e_n\}$. Let $H_j(L) = \Pr[X_j \leq L - \epsilon]$. Given an observation $\psi = (S, L)$ and a realization $\phi$ that is consistent, define the surrogate function $f$ as:

$$f(S, \phi) = n - \sum_{e_j \notin S} H_j(L) = |S| + \sum_{e_j \notin S} (1 - H_j(L)). \tag{2}$$

If no edges are probed, we set $L = \infty$, so that $\sum_{e_j \notin S} H_j(L) = n$, and $f(\emptyset, \phi) = 0$ for all $\phi$. Further, note that $f(E, \phi) = n$ for all $\phi$. Also, we clearly have that $f(S, \phi)$ depends only on the values of $\phi(S)$, and thus $f$ satisfies the self-certifying property. We now show that it also satisfies Definitions 2.3 and 2.4.

**Theorem 2.6.** *The function $f$ in (2) is strongly adaptive monotone and strongly adaptive submodular.*

*Proof.* We first show submodularity. Let $\psi_0 = (S_0, L_0)$ and $\psi_1 = (S_1, L_1)$ be two observations such that $\psi_0 \preceq \psi_1$. Thus $S_0 \subseteq S_1$ and $L_0 \geq L_1$. Consider some edge $e_k \in E \setminus S_1$. Then, noting that $f(S_0, \phi)$ depends only on $\psi_0$ and not on the realized lengths of edges not in $S_0$, we have:

$$\begin{aligned}
\Delta(k|\psi_0; \psi_1) &= \left(n - \sum_\ell g_k(\ell) \sum_{e_j \notin S_0 \cup \{e_k\}} H_j(\min(L_0, \ell))\right) - \left(n - \sum_{e_j \notin S_0} H_j(L_0)\right) \\
&= H_k(L_0) + \sum_{\ell \leq L_0} g_k(\ell) \left(\sum_{e_j \notin S_0 \cup \{e_k\}} (H_j(L_0) - H_j(\ell))\right)
\end{aligned}$$

Similarly, we have: $\Delta(\psi_1, k; \psi_1) = H_k(L_1) + \sum_{\ell \leq L_1} g_k(\ell) \left(\sum_{e_j \notin S_1 \cup \{e_k\}} (H_j(L_1) - H_j(\ell))\right)$.

Since $S_0 \subseteq S_1$, the latter summation is over fewer terms $j$. Similarly, since $L_1 \leq L_0$, the latter summation is also over fewer terms $\ell$. Since for $L_1 \leq L_0$, we have $H_j(L_1) \leq H_j(L_0)$, each term in the latter summation is also smaller. Therefore, we have $\Delta(\psi_0, k; \psi_1) \geq \Delta(\psi_1, k; \psi_1)$ showing strong adaptive submodularity.

To show monotonicity, consider an observation $\psi = (S, L)$ and let $e \notin S$ and $\ell \in \Lambda$. Define $\psi' = \psi \cup \{(e, \ell)\}$. Then we have

$$\mathbb{E}_\phi[f(S, \phi) \mid \psi \sim \phi] = n - \sum_{e_j \notin S} H_j(L),$$

$$\mathbb{E}_\phi[f(S \cup \{e\}, \phi) \mid \psi' \sim \phi] = n - \sum_{e_j \notin S, e_j \neq e} H_j(\min(L, \ell)).$$

The latter summation is over one less edge, and moreover, $H_j(\min(L, \ell)) \leq H_j(L)$. Therefore, the latter summation is at most as large as the former, which implies the condition in Definition 2.3. $\square$

### 2.4 The GREEDY Algorithm

We now show how to use the surrogate $f$ above to prove Theorem 2.1. We start with the following claim that helps us relate the success criterion in the algorithm with $f$ defined in (2).

**Claim 1** (Proved in Appendix B in supplementary material)**.** *Let $\delta \leq 1/2$ and let $\psi = (S, L)$ be an observation. For any $S, L$, we have $\Pr[\min_{e_j \notin S} X_j \leq L - \epsilon] \leq \sum_{e_j \notin S} H_j(L)$. Moreover, if $\Pr[\min_{e_j \notin S} X_j \leq L - \epsilon] \leq \delta$, then $\sum_{e_j \notin S} H_j(L) \leq 2\delta$.*

We now show how to use Theorem 2.5 to prove our main result, Theorem 2.1. To apply Theorem 2.5 directly, we introduce the following discretization. Define $\hat{H}_j(L)$ to be $H_j(L)$, rounded down to the nearest multiple of $\frac{\delta}{n}$, where $\delta$ is the parameter from Theorem 2.1. Define $Q = n - 3\delta$, and let

$$\hat{f}(S, \phi) = \min \left\{ Q, \; n - \sum_{e_j \notin S} \hat{H}_j(L) \right\}, \quad \text{where } L = \min_{e \in S} \phi(e) \text{ as before.}$$

Note that whenever $\hat{f}(S, \phi) \leq Q$, we have $|\hat{f}(S, \phi) - f(S, \phi)| < \delta$ (because the error in each of the $\hat{H}$ terms is $< \delta/n$). We are now ready to state GREEDY in Algorithm 1. (Note that $\hat{f}(S, \phi)$ can be computed just using $\psi$ when checking the condition of the while loop.)

---

**Algorithm 1** Greedy probing

---

1: Input: Probability density functions $\{g_j\}_{j=1}^n$ for edges, $H_j(L)$ as defined, parameters $\epsilon, \delta$
2: $S \leftarrow \emptyset, \psi = \emptyset$
3: Define $\hat{H}_j(L)$ as $H_j(L)$ rounded down to nearest multiple of $\delta/n$, for all $L$
4: **while** $\hat{f}(S, \phi) \leq Q - \eta$ **do**
5:     Find $e_j \notin S$ that maximizes $\Delta(j|\psi; \psi)$ (where $\Delta$ measures change in $\hat{f}$)
6:     Probe the length of $e_j$ to get value $\ell$. Add $(e_j, \ell)$ to $\psi$
7: **end while**
8: Return $S$

---

We then note that the function $\hat{f}$ is also self-certifying, strongly adaptive monotone, and strongly adaptive submodular. This follows because the proofs only rely on the monotonicty of $H$ (which also holds for $\hat{H}$) and the fact that summations involve fewer terms (which continues to hold). Thus, Theorem 2.5 applies to Algorithm 1, which lets us complete the proof of Theorem 2.1 (see Appendix B in the supplementary material for the full details).

## 3  Routing in Traffic Networks: Model and Experiments

The main assumption in Section 2 was that the lengths of the parallel paths are independent. This allowed us to find the distributions $g_j(L)$ independently for different paths, by taking into account the predictions. In practice, congestion on one path causes users to take alternate paths which also tend to get congested, and this may violate independence. However, we observe that once we probe and determine the (true) length of one of the paths, the *fluctuations* in the lengths of the other are independent. We first present the model and subsequently demonstrate its validity on real data.

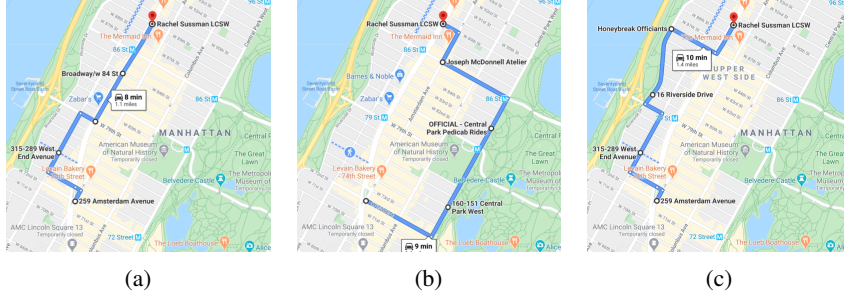

(a)                    (b)                    (c)

Figure 2: An example of three alternate routes for a query in the New York road network. Routes (b) and (c) have independent travel times after conditioning on the travel time of the canonical route (a).

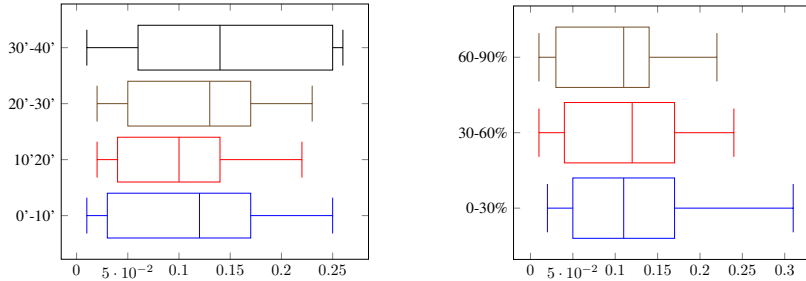

Figure 3: Pearson correlation coefficient between the lengths of two paths plotted by the length of the canonical path (left) and the overlap between them (right).

**Model for Traffic Networks.** Specifically, we designate one *canonical path*, which typically is the one that has the smallest travel time under free flow conditions (i.e., without any traffic congestion). We show that, once the length of the canonical path is probed, the lengths of the other paths have very low correlation. In other words, to a fairly good approximation, the dependencies among path lengths can be captured using a tree graphical model of depth two.

Formally, let $s, t$ be the source and destination, and let $\mathcal{P}_1, \mathcal{P}_2, \ldots, \mathcal{P}_m$ be a collection of paths, where $\mathcal{P}_1$ is the *canonical path*. Let $X_j$ be the random variable that denotes the length of $\mathcal{P}_j$. Then for any length $\ell$, the random variables $\{(X_j \mid X_1 = \ell)\}$ are all independent.

### 3.1 Experiments and Evaluation

We now use a dataset of link travel times in New York City to motivate the above model, and evaluate the performance of our greedy algorithm. This dataset contains hourly average traffic speeds on road segments throughout New York City. It covers four years of traffic estimates in New York City estimated from approximately 700 million taxi trips from 2010-2013 [Donovan and Work, 2017].

**Testing Independence Assumption.** For $1,000$ randomly sampled source-destination queries, we generate candidate paths using the plateau alternates method [Abraham et al., 2010]. For each one, we condition on the true length of the canonical path and compute the absolute value of the Pearson correlation coefficient between an arbitrary pair of (non-canonical) paths. Averaging over $1,000$ such iterations gives $0.11$ for the average absolute value of the correlation coefficient, thus providing evidence of weak correlation. Moreover, our experiments show that the coefficient remains largely constant as the canonical path length and the overlap between the paths vary. Details are given in Figure 3, and we present an example of such alternate paths in Figure 2.

**Performance of** GREEDY**.** Finally, we evaluate the GREEDY algorithm on this data. The setup is as follows. For each source-destination pair, we fix a canonical path $p_0$ and its length $\ell$ (at a certain time). For every other path $p$, the distribution $X_p$ used is the discrete, empirical distribution of its path length in all past time steps where length of $p_0$ is within $\pm 5\%$ of $\ell$. We use two different values for parameter $\delta^*$: $\delta^* = 0.1$ and $\delta^* = 0.01$. We set $\epsilon = 0$, so that we seek an exact shortest path with probability $1 - \delta^*$. We almost always recover the optimal path in the set with a small number of

probes. Specifically, with $\delta^* = 0.01$, the algorithm successfully identified the optimal path in all $1,000$ iterations, whereas with $\delta^* = 0.1$, it did so in 966 of them. For comparison, Dijkstra using historical averages recovers the optimal path in 565 of the instances. Table 1 shows the dependence of number of probes changes on the number of alternates available as $\delta^*$ changes. We note that it was not possible to identify all edge lengths with a small number of probes as, on average, $88\%$ of the candidate paths had a unique subpath. If the distributions are weakly correlated, submodularity will not necessarily hold. Our experiments also show that the greedy algorithm can be effective (though it will lose its theoretical guarantee) even when path lengths are weakly correlated.

## 4  Conclusion

In this paper, we presented a simple model and algorithm for probing for a shortest path with machine learnt advice. We validated the model assumption and algorithm on real-world data. As future work, we will study how to incorporate weaker stochastic assumptions, such as the edge independence model in Nikolova et al. [2006], as well as weaker models of expert advice than the assumption that the errors are independent and stochastic. Of particular interest is combining our results with sample complexity bounds for learning the error distributions.

| # Alternates | 10 | 20 | 30 | 40 |
|---|---|---|---|---|
| # Probes ($\delta^* = 0.01$) | 2.42 | 3.67 | 5.03 | 9.54 |
| # Probes ($\delta^* = 0.1$) | 1.75 | 2.43 | 2.95 | 4.02 |

Table 1: The average number of probes made by GREEDY for different numbers of candidate paths.

## Broader Impact

Our work has consequences to the design and implementation of algorithms in large-scale traffic routing applications. Our model is simple and easily applicable to settings where expert advice can be used to refine the choice of routes. At the same time, we make a methodological contribution by showing that for the canonical objective of probing to find the best value with high probability has a submodular surrogate that enables an efficient greedy probing strategy.

## Acknowledgments and Disclosure of Funding

Aditya Bhaskara is partially supported by NSF (CCF-2008688) and by a Google Faculty Research Award. Kamesh Munagala is supported by NSF grants CCF-1637397 and CCF-1408784; ONR award N00014-19-1-2268; and DARPA award FA8650-18-C-7880.

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
