[Supplementary Material]

# Adaptive Probing Policies for Shortest Path Routing

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

 that $f(S, \phi) = Q$ (recall that $Q$ is the the maximum value, $f(E, \phi)$). To achieve this, we need to discretize $f$. We do so by discretizing the values of $H$ (only for the purposes of computing $f$, we do not modify the distributions $g_j$). Define $\hat{H}_j(L)$ to be $H_j(L)$, rounded down to the nearest multiple of $\frac{\delta}{n}$, where $\delta$ is the parameter from Theorem 2.1. Define $Q = n - 3\delta$, and let

$$\hat{f}(S, \phi) = \min\left\{ Q, \; n - \sum_{e_j \notin S} \hat{H}_j(L) \right\}, \quad \text{where } L = \min_{e \in S} \phi(e) \text{ as before.}$$

Note that whenever $\hat{f}(S, \phi) \leq Q$, we have $|\hat{f}(S, \phi) - f(S, \phi)| < \delta$ (because the error in each of the $\hat{H}$ terms is $< \delta/n$). We are now ready to state GREEDY in Algorithm 1. (Note that $\hat{f}(S, \phi)$ can be computed just using $\psi$ when checking the condition of the while loop.)

---

**Algorithm 1** Greedy probing

---

1: Input: Probability density functions $\{g_j\}_{j=1}^n$ for edges, $H_j(L)$ as defined, parameters $\epsilon, \delta$
2: $S \leftarrow \emptyset, \psi = \emptyset$
3: Define $\hat{H}_j(L)$ as $H_j(L)$ rounded down to nearest multiple of $\delta/n$, for all $L$
4: **while** $\hat{f}(S, \phi) \leq Q - \eta$ **do**
5:     Find $e_j \notin S$ that maximizes $\Delta(j|\psi; \psi)$ (where $\Delta$ measures change in $\hat{f}$)
6:     Probe the length of $e_j$ to get value $\ell$. Add $(e_j, \ell)$ to $\psi$
7: **end while**
8: Return $S$

---

Finally, using the above machinery, we complete the analysis by proving Theorem 2.1 in Appendix B in supplementary material.

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

**Competing with the best predictor.** If none of the predictors give useful information about the true lengths $L_j$, it is clear that one needs to probe $\Omega(n)$ edges in order to find the shortest one. Thus, we need to assume that at least one of the predictors has small error. However, this is not sufficient, as the following example shows.

**Example 2.** *Suppose $n = m$, and suppose that the $i$th expert predicts length $0$ for the $i$th edge and length $1$ for all the other edges. I.e., $P_{ii} = 0$ and $P_{ij} = 1$ for all $i \neq j$. Suppose that the true lengths are $L_{j^*} = 0$ for some unknown $j^*$ and $L_j = 1$ for the rest of the edges.*

In the example above, the $j^*$th expert has a perfect prediction for every edge. But without making $\Omega(n)$ queries, it is impossible for any (potentially randomized) algorithm to find $j^*$, or equivalently, to achieve any non-trivial additive or multiplicative guarantee.

**Error or the best edge ($\gamma$).** Intuitively, the example above is difficult because there are many experts that have a large prediction error on the *best edge* (in hindsight). Consider the quantity

$$\gamma := \max_{i \in [m]} |P_{ij^*} - L_{j^*}|.$$

Even in the case of a single expert, the quantity $\gamma$ is a lower bound on the additive error that any algorithm making $o(n)$ queries can achieve. To see this, suppose we only have one expert ($m = 1$), and suppose that the predictions for all the edges is $1$, and $\gamma = 1$. Suppose that the true lengths are $1$ for all the edges except $j^*$, for which $L_{j^*} = 0$. Finding $j^*$ clearly requires $\Omega(n)$ queries.

One may hope that if we have more experts, the presence of experts who have small error on $j^*$ can help identifying $j^*$. But the following simple example shows that this is not possible.

**Example 3.** *Suppse expert $1$ predicts $P_{1j} = 1$ for all $j$, and suppose that the other experts predict $\{0, 1\}$ uniformly at random for every edge. The true lengths are $L_{j^*} = 0$ for some unknown $j^*$ and $L_j = 1$ for all other $j$.*

The same $\Omega(n)$ probe bound applies to this example, even though many (roughly $m/2$) experts predict the right value for $L_{j^*}$.

**Error of best expert.** A second quantity we consider is

$$\Delta := \min_i \max_{j \in [n]} |P_{ij} - L_j|.$$

$\Delta$ being small corresponds to the existence of an expert whose error on *all* the edges is small. While this is a strong assumption, we see that it is necessary. Even if $m = 1$ (single expert), the following example illustrates that achieving an error smaller than $\Delta$ requires $\Omega(n)$ queries.

**Example 4.** *Suppose expert $1$ predicts $P_{1j} = 0$ for $1 \leq j \leq n/2$, and predicts $P_{1j} = 1$ for $j > n/2$. Suppose that the true lengths are all $1$ except for $L_{j^*} = 0$ for some unknown $j^* \leq n/2$.*

Here, the predictions are perfect on half the edges, but we still need $\Omega(n)$ queries to find $j^*$ (i.e., to find the best edge up to an error $< \Delta(= 1)$).

*Remark.* The examples show that $\gamma$ and $\Delta$ are both necessary terms in the error bound of any algorithm with a sub-linear number of queries. The two terms are qualitatively different: $\gamma$ captures the error incurred because the predictors "over-predicted" the cost on the optimal edge, while $\Delta$ captures the error incurred because the predictors under-predicted the cost of non-optimal edges.

If we allow both the terms in the error bound, we do not need any queries to achieve a good routing.

**Routing without probes.** Consider simply using the edge $j$ that has the minimum value of $\max_i P_{ij}$. For the best edge $j^*$, this quantity is $\leq L_{j^*} + \gamma$, by assumption. Suppose $j$ is the edge that we output. Since there exists an expert $i^*$ whose error on that edge is $\leq \Delta$, we have that $L_j \leq P_{i^*j} + \Delta \leq L_{j^*} + \gamma + \Delta$.

The examples and the algorithm above show that probing does not offer much power when predictions can be arbitrary. Moreover, there are limitations on how close we can get to the optimal edge length.

# B    Omitted Proofs

## B.1    Proof of Claim 1

The first part of the claim follows by a simple union bound:

$$\sum_{e_j \notin S} H_j(L) = \sum_{e_j \notin S} \Pr[X_j \leq L - \epsilon] \geq \Pr\left[\cup_{e_j \notin S}(X_j \leq L - \epsilon)\right] = \Pr[\min_{e_j \notin S} X_j \leq L - \epsilon]$$

To see the second part, note that since $H_j(L) = \Pr[X_j \leq L - \epsilon]$,

$$\delta = \Pr[\min_{e_j \notin S} X_j \leq L - \epsilon] = 1 - \prod_{e_j \notin S}(1 - H_j(L)) \geq 1 - e^{-\sum_{e_j \notin S} H_j(L)}.$$

Therefore,

$$e^{-\sum_{e_j \notin S} H_j(L)} \geq 1 - \delta \implies \sum_{e_j \notin S} H_j(L) \leq -\ln(1 - \delta) \leq 2\delta,$$

where the final inequality holds when $\delta \leq 1/2$.

## B.2    Proof of Theorem 2.1

We begin by noting that Claim 1 lets us relate a policy that maximizes $\hat{f}$ to an edge probing policy that succeeds with parameters $(\epsilon, \delta)$, as defined in Eq. (1), and vice versa. Consider any edge probing policy $\pi$ that succeeds with parameters $(\epsilon, \delta)$. In other words, whenever the procedure terminates, we have a set of edges $S$ with minimum edge length $L$ that satisfies (1). Using Claim 1, this implies that $f(S, \phi) \geq n - 2\delta$ for all realizations $\phi$ consistent with the edge observations. Thus, we have $\hat{f}(S, \phi) = Q$ when the procedure terminates. I.e., $\pi$ is also a valid policy for maximizing $\hat{f}$ (in the sense of Theorem 2.5).

Conversely, consider a policy that aims to maximize $\hat{f}$ and achieve $\hat{f} > Q - \eta$. It terminates when $\hat{f}(S, \phi) = Q$, because of our discretization. This implies that $f(S, \phi) \geq n - 4\delta$, or that $\sum_{e_j \notin S} H_j(L) \leq 4\delta$. Using Claim 1, this means that the policy succeeds with parameters $(\epsilon, 4\delta)$ for edge probing.

Next, we note that the function $\hat{f}$ is also self-cerifying, strongly adaptive monotone, and strongly adaptive submodular. This follows because the proofs only relied on the monotonicty of $H$ (which also holds for $\hat{H}$) and the fact that summations involve fewer terms (which continues to hold). Thus, Theorem 2.5 applies to Algorithm 1, and using the above connection between policies for edge probing and policies for maximizing $\hat{f}$, we conclude that Algorithm 1 succeeds with parameters $(\epsilon, 4\delta)$, and that the expected number of queries is OPT $\cdot O\left(\log\left(\frac{Q}{\eta}\right) + 1\right)^2$. This completes the proof.