[Reviews · NeurIPS 2020]

Review 1

Summary and Contributions: This work considers the problem of shortest path routing with an expensive oracle. The goal is an adaptive policy for choosing which paths to ask for help on rather than using lower quality estimates of edge/path delays coming from multiple potential machine learning models. The paper shows how to fit this problem into the the adaptive submodularity framework from Golovin & Krause, which then certifies that a greedy algorithm comes within a log factor of the number of queries in any optimal approach. While the theoretical model is limited to parallel edges, real data is used to show that the policy still behaves well on a realistic network (NYC), and is able to consistently find the best paths.

Strengths: This work shows a nice relation between the theoretical framework of adaptive submodular maximization and the very applied problem of querying expensive oracles for assistance with routing. The theoretical reduction is further backed up with experimental data, which show that the approach is able to find the best paths almost all the time with a small number of queries. This manner of reduction may be of significant interest to other settings in which there is some amount of offline data e.g. on a phone, and much better data on a central server, and this paper did a good job of illustrating the reduction.

Weaknesses: The overall significance of the work is somewhat limited as an application of the existing adaptive submodular maximization framework to a very interesting setting. As the theoretical result is about a comparison to OPT, it would be nice (though a challenge) to be able to include within the analysis of GREEDY an optimal or near optimal policy. Where is the gap? Since the numbers of probes are so small it can't be too far, but it would be interesting to see.

Correctness: The claims and methods seem to be correct, though I have not verified proofs.

Clarity: The paper is very well written, and nicely formulates the problem, the framework, reduction and experimental results.

Relation to Prior Work: This work clearly discusses how it differs from and relates to previous contributions.

Reproducibility: Yes

Additional Feedback:


Review 2

Summary and Contributions: The paper addresses a problem in traffic routing. It considers a theoretical problem involving n independent paths between s and t, where there is a known probability distribution on the length of each path, and the actual length of a path can be determined by probing the edge. Probing must continue until a path can be identified that is at most epsilon longer than the shortest path, with probability delta. An approximation algorithm for this problem is presented, which works by constructing a submodular surrogate function and running the Adaptive Greedy algorithm of Golovin and Krause. Using New York City taxi data, the probabilistic model is verified and the model is presented.

Strengths: Traffic routing is an important and well-studied problem. This paper presents a new and clean theoretical version of the routing problem which is interesting and natural. The presented approximation algorithm is a nice application of submodular surrogates and Adaptive Greedy. The experimental section, with its model justification and experiments on real-world data, make the paper a good mix of theory and practice.

Weaknesses: [I modified this answer after reading the author feedback. I was happy with their responses to my original questions/concerns] None, except that the paper needs some minor corrections and changes. See below under Additional Feedback.

Correctness: Yes (new answer after reading author feedback)

Clarity: Yes.

Relation to Prior Work: Yes.

Reproducibility: Yes

Additional Feedback: [I updated this answer after reading the author feedback.] In my original review, I asked about the distribution used in the experiments. The authors gave a good explanation of the distribution in their feedback and said they would incorporate it into the final version. I'm happy with this. The submodularity and monotonicity conditions are proved for the continuous function f, but the algorithm is run on the discretized version of f, which is \hat{f}. The authors modify these proofs to apply to the discretized version of f (which is a trivial modification). Together with that change, the authors need to be careful in what follows (both in the main paper and the supplementary material) to specify when they are talking about f vs. \hat{f}, and when they are talking about H vs. \hat{H}. The way things are written now is somewhat confusing, because this distinction was not made clearly. There are two typos in the statement of Theorem 2.1: 3\delta should be changed to 4\delta, and the given approximation bound needs to be squared.


Review 3

Summary and Contributions: Reduces a traffic routing problem - finding fastest route through network where journey times are constantly changing - to a simpler graph problem where all possible routes are represented as single, independent edges from start to sink. Then tests the resultant model against real data. Likely journey times are represented as distributions, as drawn from multiple ML models. Accurate journey times are available by probing a traffic server. The objective is to find a search policy that minimises the number of probes. The paper makes two contributions: the search policy adapted from Golovin and Krause and an empirical demonstration that the graph simplification provides a good enough approximation of real data.

Strengths: Theoretical claims supported by experimentation. Problem is contained and clearly articulated. The appendix is valuable and its content well-signposted in the main text.

Weaknesses: The authors are up-front about their assumption that edge lengths are independent even though, in practice, they may not be but then make a claim which is described as "a fairly good approximation" but muddied by being given formally, as if it were proved somewhere. It feels like a bit of a gap in an otherwise compelling story.

Correctness: I believe so. See comments at 8. Also, I would flag in place that the proof to Theorem 2.1 is discussed later with full proof in the supplementary material. When you say at l290 that you will 'show' that other paths have low correlation with the length of the canonical path, I expected a proof. Especially given the last para of that intro to 3,which looks very like a theorem. Perhaps make explicit that you're going to show it experimentally. Is this a property that you hoped to prove theoretically?

Clarity: Yes, very. Clear and succinct, well-structured. Some minor stylistic inconsistencies, e.g., definition in round brackets at 5.212 then in square brackets and italics at 221.

Relation to Prior Work: I like that it is clear where definitions have been borrowed from previous work - but conveniently included where they're needed.

Reproducibility: Yes

Additional Feedback: Could be clearer in the intro that typically it isn't the lengths of the paths that change in a traffic network but the time taken to traverse those paths. p2 when you italicise a term - e.g., 'adaptive' - it would be useful to follow up with the way you're defining it (a "that is, blah blah") - otherwise I'm left wondering "what did they mean by that?" I assume that, if you're not using those terms in a very particular way, they don't need italicising. There's something confusing in the description of the main contribution that what's given as "probability" is actually "probability not". I don't know what the solution is but maybe explicitly call it probability of error. Or find a positive term for it. g_j is only defined as the density function after its first use - in the unnumbered equation p4 l160. 'r.v.s.' at the end of a sentence is a bit unfortunate because it looks like a novel abbreviation Have you lost half a sentence at p4. l174 - "We say that a policy succeeds with parameters (\epsilon, \delta) if at any termination...? I would suggest numbering the self-certifying property as a formal definition since it's depended on for Theorem 2.5 and you later reference it in comparison to submodularity. ========== Thanks to the authors for their rebuttal and to my fellow reviewers for the discussion.


Review 4

Summary and Contributions: The authors considered a problem where the goal is to find the minimum value among a random realization of a set of independent random variables with known distributions. The problem is related to stochastic probing but not exactly the same. Specifically, the authors were considering the probability that the smallest random variable is not probed. A few theoretical results were presented, including an adaptive submodular proxy of the probability considered. With this proxy function, the authors showed that a greeding probing scheme is optimal within a constant gap, using results from the paper [Adaptive Submodularity: Theory and Applications in Active Learning and Stochastic Optimization].

Strengths: The paper is motivated by a real-world example -- traffic routing. The paper abstracted it as adaptive learning problem, and discussed some theoretical results about it. The overall presentation flows naturally and the logic is relatively easy to follow. The theoretical results can be potentially useful to solve practical problems.

Weaknesses: Although I understand that the traffic-routing example serves as a motivation, it seems that it is not as strongly connected to the problem as it seems to be. The traffic-routing objective, as is, may be more related to the stochastic probing model. In reality, if the second-best route is only slightly slower, then having chosen that route should ideally incur a small penalty. While in the paper's setting, it incurs the same penalty as having chosen the worst route, which seems unrealistic. The relaxation into an adaptive-submodular proxy (section 2.3) is a nice idea, but I do have trouble piecing the proofs together. The discussions about the proof of theorem 2.1 (line 267-275) is quite confusing. Specifically, Q is the max possible value, and \delta is a chosen slackness in theorem 2.1. As such line 275 will not go through -- $Q=n-3\delta$ will not hold in general as all the numbers are given. The evaluation section is missing several details. I appreciate that the authors empirically checked the correlation between alternative routes but it's still unclear to me what the implications are for weakly-correlated random variable probing -- submodularity will no longer hold. In line 309 and below it's unclear what distributions were used in the evaluation -- i.e. what are the assumptions on the "expert hints" that lead to the results. Like whether the distributions are "friendly" (e.g. means are different and tails are thin) or not. I understand that the paper has a page limit, but I think several additional lines will help clarify things out.

Correctness: I am not sure the proof is correct, or if I misunderstood some part of the technical details. I think the empirical methodologies are on the right track but some more details are going to be helpful.

Clarity: The paper is well written. I like the fact that it is well-organized, with clear motivation, related work, the abstracted theoretical question and analysis. I find it relatively easy to follow.

Relation to Prior Work: The related work are clearly presented and well organized into three categories. The authors clearly stated the uniqueness of their work when comparing with previous research.

Reproducibility: Yes

Additional Feedback: This is to acknowledge that I have read the rebuttal. I'm still not quite convinced about whether \epsilon will fully capture the "bad vs almost best" effect but I think it does not matter as much, since it is a deliberate design choice in the problem formulation.

[Author Response · NeurIPS 2020]

We thank the reviewers for their insightful comments. We will fix the typos and make the writing improvements suggested by the reviewers in the final version. Below, we address the main questions raised by the reviewers.

**Reviewers 2 and 4: Correctness.** We are confident the analysis is correct, and address the reviewer concerns below.

- *Finiteness of state space:* We assume the distributions are discrete on line 166, and the number of distributions is finite. Therefore the state space is finite. To avoid confusion, we will omit the claim that all our arguments carry over to continuous distributions, and add "finite support" for each distribution.

- *Discretization:* The proof that the discretized function satisfies the preconditions of Golovin-Krause is in the Supplementary material in Appendix B.2, Proof of Theorem 2.1 (lines 473-476 of the supplement). We will include it in the main paper in the final version.

- *Paragraph of line 270.* The notation $f, Q$ used in lines 270–272 is from the statement of Theorem 2.5, and not the same as the $f, Q$ used subsequently in the paragraph. We will delete the sentences "To apply Theorem ... To achieve this,..." in lines 270–272. In the rest of the paragraph, Theorem 2.5 and the Greedy algorithm are applied to the function $\hat{f}$ whose maximum value is $Q = n - 3\delta$, and for which $\eta$ as defined in Theorem 2.5 is equal to $\frac{\delta}{n}$. Thanks for catching this. We will clarify this point and fix the writing.

**Reviewers 2 and 4. Distribution in the Experiments.** For running Greedy, we fix a canonical path $p_0$ and its length $\ell$ at the current time. For each other path $p$, the distribution $X_p$ used is the discrete, empirical distribution of its path length in all past time steps where length of $p_0$ is within $\pm 5\%$ of $\ell$. For testing the independence assumption, we use all steps instead of all past time steps. We will add these details to the make the exposition clear.

Our goal is to evaluate the performance of the greedy algorithm from Section 2, therefore, we assumed known distributions, and we did not experiment with how these distributions are generated from expert hints.

If the distributions are weakly correlated, submodularity will not necessarily hold. The goal of the experiments is to show that the algorithm itself is still empirically effective (though it will lose its theoretical guarantee). We will clarify this point.

**Reviewer 4, Penalty for bad routes vs. "nearly best" routes.** The parameter $\epsilon$ in our discussion captures exactly this slack. In Equation (1), we are assuming that routes whose length is within a given $\epsilon$ of the best route are also acceptable. Therefore, we are implicitly giving a low penalty to nearly-best routes, and our goal is to find such a route with high probability. We will highlight this better and earlier in the text.

[Meta-Review · NeurIPS 2020]

This paper gives an adaptive policy for choosing the shortest path between two given nodes in a network when the designer knows the distribution of the lengths but each edge-probe is expensive. The algorithm is based on a neat reduction to adaptive submodular optimization framework. The reviewers had a substantial discussion on the merits and assumptions of this work and were convinced that it gives a neat result on a elegant and important problem. I am pleased to recommend acceptance.